# CONTROLVAR: EXPLORING CONTROLLABLE VISUAL AUTOREGRESSIVE MODELING

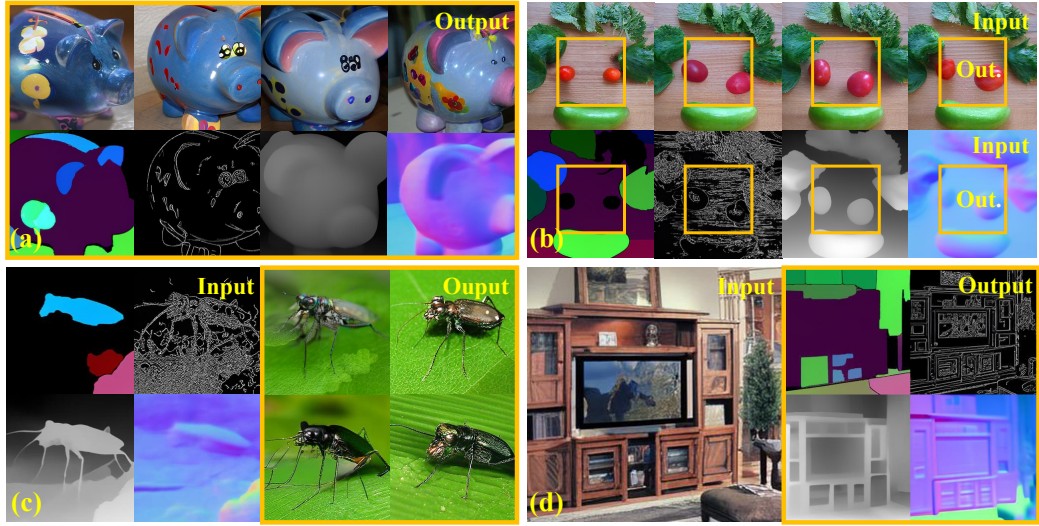

Figure 1: Visualization of ControlVAR for (a) joint control-image generation, (b) joint control-image completion, (c) control-to-image generation, and (d) image-to-control prediction (visual perception tasks). The yellow boxes denote the predicted images/controls.

## ABSTRACT

Conditional visual generation has witnessed remarkable progress with the advent of diffusion models (DMs), especially in tasks like control-to-image generation. However, challenges such as expensive computational cost, high inference latency, and difficulties of integration with large language models (LLMs) have necessitated exploring alternatives to DMs. This paper introduces ControlVAR, a novel framework that explores pixel-level controls in visual autoregressive (VAR) modeling for flexible and efficient conditional generation. In contrast to traditional conditional models that learn the conditional distribution, ControlVAR jointly models the distribution of image and pixel-level conditions during training and imposes conditional controls during testing. To enhance the joint modeling, we adopt the next-scale AR prediction paradigm and unify control and image representations. A teacher-forcing guidance strategy is proposed to further facilitate controllable generation with joint modeling. Extensive experiments demonstrate the superior efficacy and flexibility of ControlVAR across various conditional generation tasks against popular conditional DMs, e.g., ControlNet and T2I-Adaptor.

## 1 INTRODUCTION

In recent years, conditional image generation Zhang et al. (2023); Mou et al. (2023); Esser et al. (2021); Tian et al. (2024); Nam et al. (2024) has attracted great attention and there have been significant advancements in text-to-image generation Rombach et al. (2021); Chang et al. (2023); Gal et al. (2022), image-to-image generation Zhang et al. (2023); Mou et al. (2023); Ruiz et al. (2023),

(a) Pixel-conditioned next-token prediction      (b) Controllable next-scale prediction

Figure 2: In contrast to previous methods Esser et al. (2021); Zhan et al. (2022) that leverage prefix conditional tokens to impose controls, ControlVAR jointly models the pixel-level controls and image during training and conducts the conditional generation tasks during testing with the teacher forcing. Class and type tokens provide semantic and control type (mask, canny, depth and normal) information respectively.

and even more complex tasks Nam et al. (2024); Li et al. (2023b; 2024). Most recent approaches, e.g., ControlNet Zhang et al. (2023), leverage the powerful diffusion models (DMs) Rombach et al. (2021); Peebles & Xie (2023) to model the large-scale image distribution and incorporate additional controls with classifier-free guidance Ho & Salimans (2022). However, the inherent nature of the diffusion process imposes many challenges for the diffusion-based visual generation: (1) the computational cost and inference time are significant due to the iterative diffusion steps Song et al. (2020a); Ho et al. (2020) and (2) the incorporation in mainstream intelligent systems, i.e., large language models (LLMs) Touvron et al. (2023); Achiam et al. (2023), is intricate due to the representation difference. This motivates the community to find a replacement for DMs for high-quality and efficient visual generation in the era of LLMs.

Inspired by the success of autoregressive (AR) language modeling Touvron et al. (2023); Achiam et al. (2023), AR visual modeling Esser et al. (2021); Tian et al. (2024) has been investigated as a counterpart to DMs given its strong scalability and generalizability Tian et al. (2024); Bai et al. (2023). Several inspiring works, e.g., VQGAN Esser et al. (2021), DALL-E Ramesh et al. (2021a) and VAR Tian et al. (2024), have demonstrated promising image generation results with AR modeling. Nevertheless, compared to the prosperity of conditional DMs Zhang et al. (2023); Mou et al. (2023); Chen et al. (2022); Xu et al. (2023); Qin et al. (2023); Ju et al. (2023), visual generation with conditional AR modeling Zhan et al. (2022); Esser et al. (2021) remains significantly under-explored. Different from DMs, where all the pixels are modeled simultaneously, AR models are characterized by modeling sequential values based on their corresponding previous ones. This AR approach naturally leads to a conditional model, providing potential flexibility when incorporating additional controls. To leverage this property, teacher forcing is a popular approach that controls AR prediction by replacing partially predicted tokens with ground truth ones Esser et al. (2021). Thanks to this nature of AR modeling, we found that highly flexible conditional generation can be achieved by teacher forcing partial AR sequence with proper model designs.

In this paper, we explore the **Control**lable **V**isual **A**utoreg**R**essive modeling with both token-level and pixel-level conditions. A new conditional AR paradigm, ControlVAR is introduced, which permits a highly flexible conditional image generation by embracing the next-scale prediction of joint control and image (Fig. 2(b)). Previous wisdom Zhan et al. (2022); Esser et al. (2021) typically utilizes prefix conditions (Fig. 2(a)) and mainly model images from raw pixel space in an AR manner. Differently, we notice that if we jointly model the control and image, the learned joint prediction can be easily guided by teacher forcing during inference. On the one hand, we unify the control and image representations and reformulate the sequential variables for the AR process to enable effective joint modeling. On the other hand, by analyzing the modeled probabilities, we introduce an effective sampling strategy, named teacher forcing guidance (TFG) to facilitate conditional sampling. Remarkably, a single ControlVAR model trained via TFG is capable of multiple meaningful tasks with different input-output combinations between control and image: (a) joint control-image generation, (b) control/image completion, (c) control-to-image generation, (d) image-to-control generation, as demonstrated in Fig. 1. Beyond the image-control tasks that are jointly modeled during training,

we observe that ControlVAR also emerges capabilities for unseen tasks, e.g., control-to-control generation, further enhancing its flexibility and versatility. Our contribution can be summarized in three-fold:

- We present ControlVAR, a novel framework for controllable autoregressive image generation with strong flexibility for heterogeneous conditional generation tasks.

- We unify the image and control representations and reformulate the conditional generation process to jointly model the image and control during training. To perform conditional generation during inference, we introduce teacher-forcing guidance (TFG) that enables controllable sampling.

- We conduct comprehensive experiments to investigate the impacts of each component of ControlVAR and demonstrate that ControlVAR outperforms powerful DMs methods, e.g., ControlNet and T2I-Adapter on controlled image generation across several pixel-level controls, i.e., mask, canny, depth and normal.

## 2 RELATED WORKS

### 2.1 DIFFUSION-BASED IMAGE GENERATION

The evolution of diffusion models, initially introduced by Sohl-Dickstein et al. Sohl-Dickstein et al. (2015) and later expanded into image generation using fixed Gaussian noise diffusion processes Ho et al. (2020); Song et al. (2020b), has witnessed significant advancements driven by various research efforts. Nichol et al. Nichol & Dhariwal (2021) and Dhariwal et al. Dhariwal & Nichol (2021) proposed techniques to enhance the effectiveness and efficiency of diffusion models, paving the way for improved image generation capabilities. Notably, the paradigm shift towards modeling the diffusion process in the latent space of pre-trained image encoders as a strong prior Van Den Oord et al. (2017); Esser et al. (2021) rather than raw pixels spaces Vahdat et al. (2021); Rombach et al. (2022); Peebles & Xie (2023) has been instrumental in achieving high-quality image generation with reasonable inference speed. This approach has led to the development of foundational diffusion models such as Glide Nichol et al. (2021), Cogview Ding et al. (2021; 2022); Zheng et al. (2024), Make-a-scene Gafni et al. (2022b), Imagen Saharia et al. (2022), DALL.E Ramesh et al. (2021b), Stable Diffusion Stability AI (2022), MidJourney MidJourney Inc. (2022), SORA OpenAI (2024), among others, which are often pre-trained on large-scale data with conditions, typically text Gordon et al. (2023); Webster et al. (2023); Elazar et al. (2023); Chen et al. (2024). Recent advancements include consistency models derived from diffusion models Song et al. (2023); Song & Dhariwal (2023); Luo et al. (2023), enabling generation with reduced inference steps. These foundational diffusion models have not only opened doors to novel downstream applications like Text inversion Gal et al. (2022), DreamBooth Ruiz et al. (2023), T2I-Adapter Mou et al. (2023), ControlNet Zhang et al. (2023), but also inspired a plethora of research in controllable generation Meng et al. (2021); Brooks et al. (2023); Huang et al. (2023d); Tumanyan et al. (2023); Voynov et al. (2023); Huang et al. (2024; 2023a); Bashkirova et al. (2023); Bar-Tal et al. (2023); Li et al. (2023c); Qi et al. (2023); Zhan et al. (2022) and other innovative areas.

### 2.2 AUTOREGRESSIVE IMAGE GENERATION.

Unlike diffusion-based models that typically leverage continuous image representation, autoregressive models Huang et al. (2023c); Esser et al. (2021); Van den Oord et al. (2016); Tian et al. (2024) leverage discrete image tokens. An image tokenizer Esser et al. (2021); Yu et al. (2023; 2024); Huang et al. (2023b); Ge et al. (2023) is utilized to encode the image into a sequence of discrete tokens. VQGAN Esser et al. (2021) first patches the image and then employs a vector-quantization approach to discretize the image features. Following this paradigm, a series of following-up works improve the image tokenization by using more powerful quantization operations Huang et al. (2023c); Lee et al. (2022); Yu et al. (2023), reformulating the image representation Tian et al. (2024); Tschannen et al. (2023) and modifying the network architecture Yu et al. (2021); Razavi et al. (2019). With the discrete tokens, a transformer structure Radford et al. (2019) is leveraged to model the image token sequences. RQ-GAN Lee et al. (2022) improves the modeling by incorporating a hierarchy design and MQ-VAE Huang et al. (2023c) further utilizes StackTransformer to enhance the spatial

focus. MUSE Chang et al. (2023) is a large-scale pre-trained text-to-image model where a low-resolution image is first generated followed by a super-resolution transformer to refine the image. Recently, VAR Tian et al. (2024) introduced a new next-scale autoregressive prediction paradigm where the image representation is shifted from patch to scales. The new representation is featured with the maintenance of spatial locality and much lower computational cost. In this paper, we follow the next-scale autoregressive paradigm and explore the incorporation of additional controls into the modeling process.

## 2.3 CONDITIONAL IMAGE GENERATION

Though significant progress has been made in generating highly realistic images from textual descriptions, describing every intricate detail of an image solely through text poses challenges. To overcome this limitation, researchers have explored alternative approaches using various additional inputs to effectively control image and video diffusion models. These inputs encompass bounding boxes Li et al. (2023c); Yang et al. (2023), reference object images Ruiz et al. (2023); Li et al. (2023a), segmentation maps Gafni et al. (2022a); Avrahami et al. (2023); Zhang et al. (2023), sketches Zhang et al. (2023), and combinations thereof Kim et al. (2023); Qin et al. (2023); Zhao et al. (2024); Wang et al. (2024); Mizrahi et al. (2024); Nam et al. (2024); Zhou et al. (2023). However, fine-tuning the vast array of parameters in these diffusion models can be computationally intensive. To address this, methods like ControlNet Zhang et al. (2023) have emerged, enabling conditional control through parameter-efficient training strategies Zhang et al. (2023); Ryu (2022); Mou et al. (2023). Notably, X-Adapter Ran et al. (2024) innovatively learns an adapter module to adapt ControlNets pre-trained on smaller image diffusion models (e.g., SDv1.5) for larger models (e.g., SDXL). SparseCtrl Guo et al. (2023) takes a different approach, guiding video diffusion models with sparse conditional inputs, such as few frames instead of full frames, to mitigate the data collection costs associated with video conditions. However, the implementation of SparseCtrl necessitates training a new variant of ControlNet from scratch, as it involves augmenting ControlNet with an additional channel for frame masks. Beyond traditional conditional image generation, the in-context learning capability of conditional models has also been explored Safaee et al. (2023); Mizrahi et al. (2024); Bai et al. (2023); Zhang et al. (2024). LVM Bai et al. (2023) investigates the scaling learning capability of a large vision model without any linguistic data. 4M Mizrahi et al. (2024) investigate the large-scale visual generation with multimodal data using masked image modeling. Different from previous works which are mainly focusing on diffusion models, we aim to explore adding additional control to the autoregressive visual generation process.

## 3 CONTROLVAR

ControlVAR is an autoregressive Transformer Vaswani et al. (2017) framework for conditional image generation tasks, using the following as conditions: image $I \in \mathbb{R}^{3 \times H \times W}$, pixel-level control $C \in \mathbb{R}^{3 \times H \times W}$ and token-level control $c \in \mathbb{R}^D$ where $H, W$ and $D$ denotes the image size and dimension of control token respectively. We denote the set of $N$ different types of controls as $\mathcal{C} = \{C_n\}_{n \in [N]}$.

**Problem formulation.** Prior conditional approaches Zhang et al. (2023); Tian et al. (2024) have often utilized distinct models for individual control type $C$, learning a conditional distribution in the form of $p(I|C, c)$, where each image $I$ is encoded as a sequence of discrete tokens of length $T$, denoted as $(x_1, x_2, \ldots, x_T)$. By employing autoregressive (AR) modeling, we can rewrite the conditional probability $p(I|C, c)$ as

$$p(I|C, c) = p(x_1, x_2, \ldots, x_T|C, c) = \prod_{t=1}^{T} p(x_t|x_{<t}, C, c) \tag{1}$$

where each image token $x_t$ is conditioned on previous ones $x_{<t}$ at position $t$ and prefix controls $C, c$.

In this paper, we consider $N$ different controls and reformulate the conditional AR generation to model the joint distribution $p(I, \mathcal{C}|c)$ during training. Specifically, we uniformly sample one control $C \in \mathcal{C}$ at each training iteration and leverage an additional type token $c_t$ to convey the control type

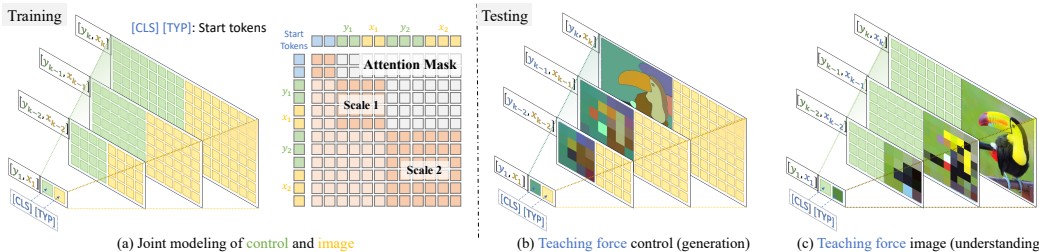

(a) Joint modeling of control and image

(b) Teaching force control (generation)

(c) Teaching force image (understanding)

Figure 4: Illustration of ControlVAR. We jointly model the control and image during training with start tokens [CLS] and [TYP] to specify the semantics and control type. We conduct conditional generation by teacher forcing the AR prediction during testing.

information. Assuming the control tokens are of the same length as the image (which we will show in the next section), we represent it as a sequence of discrete tokens $C = (y_1, y_2, \ldots, y_T)$. To jointly model the image and control while not losing the autoregressive properties, we group the image and control tokens as $r_t = (x_t, y_t)$ and model the joint distribution as:

$$p(I, C|c, c_t) = p((x_1, y_1), (x_2, y_2), \ldots, (x_T, y_T)|c, c_t) = \prod_{t=1}^{T} p(r_t|r_{<t}, c, c_t). \quad (2)$$

For inference, we introduce an innovative approach inspired by teacher forcing, which replaces the predicted token with the ground truth to perform conditional generation tasks. We will discuss the representation of $r_t$ in Sec. 3.1, joint control-image AR modeling in Sec. 3.2, and conditional generation during inference in Sec. 3.3.

### 3.1 UNIFIED IMAGE AND CONTROL REPRESENTATION.

Images are generally represented in RGB, which is different from how pixel-level controls (e.g., mask, canny, and depth) are represented. Although using the original representation of respective controls may be beneficial for information preservation, doing so would lead to a larger vocabulary size of the predicted tokens thus hindering effective AR modeling. To this end, we aim to represent the controls with the same RGB representation of images.

**Control representation.** We consider four popular control types - entity mask, canny, depth, and normal in this paper. We notice that canny, depth, and normal can be easily converted to RGB by using simple transformations Zhang et al. (2023). However, entity segmentation masks $M \in \{0, 1\}^{N \times H \times W}$ which comprises $N$ class-agnostic binary masks (Fig. 3(b)) cannot be easily converted. Inspired by SOLO Wang et al. (2020), we leverage a position-aware color map to encode the binary masks $M$ into a colormap $M' \in [0, 255]^{3 \times H \times W}$. To better distinguish the color difference, we select 5 candidate values $\{0, 64, 128, 192, 255\}$ from each RGB channel and combine them to $124 = 5^3 - 1$ colors ($(0, 0, 0)$ is preserved for background). To apply the colormap, as shown in Fig. 3, we divide the image into $n_h \times n_w$ regions where each region represents a corresponding color. We calculate the centeredness of each mask and apply the colors to masks based on their centeredness locations. Therein, we can convert the entity masks to a RGB colormap.

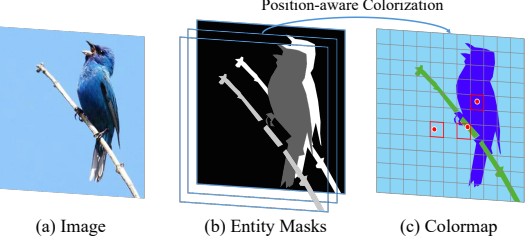

(a) Image  (b) Entity Masks  (c) Colormap

Figure 3: Illustration of colormap representation.

**Tokenization.** As the control and image share the same RGB representation, we can utilize the same approach to tokenize them. To represent an RGB image as a sequence of discrete tokens $(x_1, x_2, \ldots, x_T)$, patch-level Esser et al. (2021) and scale-level Tian et al. (2024) representations

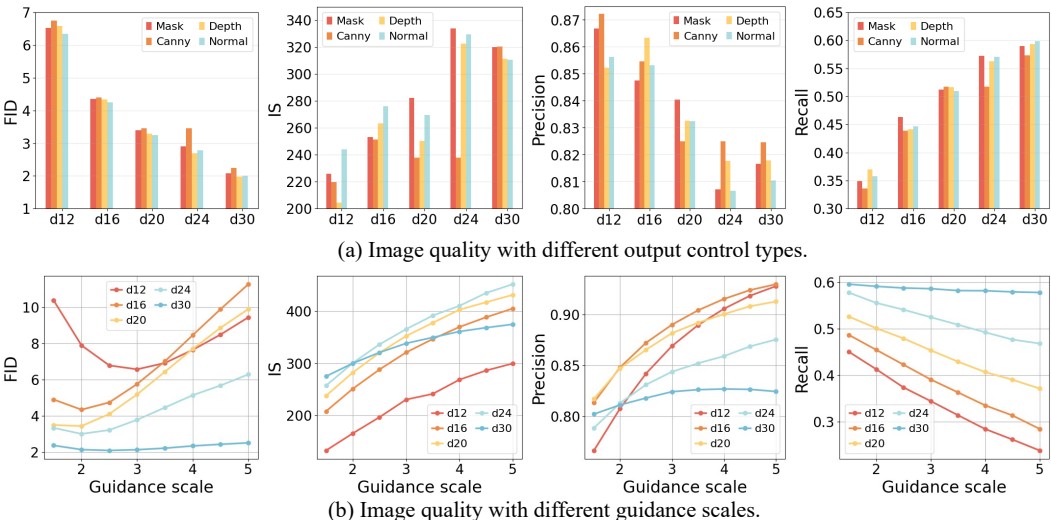

(a) Image quality with different output control types.

(b) Image quality with different guidance scales.

Figure 5: Joint control-image generation with (a) different output control types, (b) guidance scales.

have been explored. The patch-level tokenization process splits an image into $T$ patches and represents each patch as a token $x_t$ where $x_t \in [V]^1$ is an integer from a vocabulary of size $V$. Recently, a scale-level representation has been introduced which decomposes the image into $T$ scales where each scale is represented by a set of tokens $x_t \in [V]^{h_t \times w_t}$ (Fig. 2(b)). $h_t \times w_t$ denotes the size of the $t$-th scale. Compared to patch-level representation, scale-level representation can better preserve the spatial locality and capture global information which are desired for conditional image generation tasks. This motivates us to adopt the scale-level representation in our approach. Specifically, we obtain the image tokens and control tokens using the shared tokenizer $\Phi$ as

$$(x_1, x_2, \ldots, x_T) = \Phi(I), \quad (y_1, y_2, \ldots, y_T) = \Phi(C). \tag{3}$$

Here, $x_t \in [V]^{h_t \times w_t}$ and $y_t \in [V]^{h_t \times w_t}$ share the same vocabularies, which makes it easier for joint control-image AR modeling.

## 3.2 JOINT CONTROL-IMAGE MODELING

We demonstrate the network details for joint modeling in this section. Following VAR Tian et al. (2024), we leverage a GPT-2 style Transformer network architecture for our ControlVAR models. As shown in Fig. 4 (a), we jointly model the control and image in each stage. A flatten operation is adopted to convert the sequence of 2D features into 1D. Full attention is enabled for both control and image tokens belonging to the same scale, which allows the model to maintain spatial locality and to exploit the global context between control and image. A standard cross entropy loss is used to supervise our autoregressive ControlVAR models.

Specifically, we employ two pre-defined special tokens $c = [\text{CLS}] \in [N_{cls}]^1$ and $c_t = [\text{TYP}] \in [N_{typ}]^1$ as the start tokens. $N_{cls}$ and $N_{tpy}$ denote the number of classes and control types respectively. [CLS] token aims to provide semantic context for the generated image. [TYP] token is used to select the type of pixel-level control to be generated along with the image. Following previous works Chang et al. (2023); Tian et al. (2024), additional empty tokens are used to replace special tokens with a probability of $\delta$ during training to apply classifier-free guidance Ho & Salimans (2022).

## 3.3 SAMPLING WITH TEACHER-FORCING GUIDANCE.

Classifier-free guidance (CFG) Ho & Salimans (2022) was originally introduced to apply and enhance the effect of conditional controls on diffusion models without an explicit classifier. Extensive studies Sanchez et al. (2023); Chang et al. (2023); Tian et al. (2024) have demonstrated that classifier-free guidance also works for AR models.

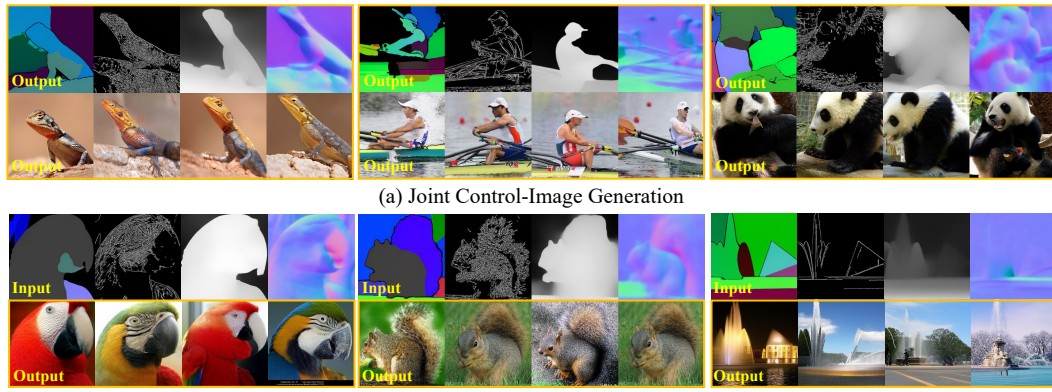

(a) Joint Control-Image Generation

(b) Control-to-Image Generation

Figure 6: Visualization of (a) joint control-image generation and (b) control-to-image generation.

Here, we analyze how to achieve conditional generation by using the image generation task $p(I|C, c, c_t)$ as an example. Given image $I$, pixel-level control $C$ and token-level controls $c, c_t$, CFG Ho & Salimans (2022) leverages Bayesian rule to rewrite the conditional distribution as

$$p(I|C, c, c_t) \propto p(c|I, C, c_t)p(c_t|I, C)p(C|I)p(I). \tag{4}$$

It can be seen that the class $c$ and control type $c_t$ are independent. By applying the Bayesian rule again, we have

$$p(c|I, C, c_t) = \frac{p(I, C|c, c_t)p(c, c_t)}{p(I, C|c_t)p(c_t)} = \frac{p(I, C|c, c_t)p(c)}{p(I, C|c_t)}. \tag{5}$$

Given the AR nature of ControlVAR, $p(I, C|c, c_t)$ and $p(I, C|c_t)$ can be induced by using the pixel-level condition $C$ to teacher-force ControlVAR during the AR prediction. Similarly, after rewriting all terms in Eq. (4) to the form in Eq. (5), we derive an approach to sample with both pixel- and token-level controls for image generation as

$$\begin{aligned} x^* = x(\vec{\emptyset}|\emptyset, \emptyset) &+ \gamma_{cls}(x(\vec{C}|c, c_t) - x(\vec{C}|\emptyset, c_t)) \\ &+ \gamma_{typ}(x(\vec{C}|\emptyset, c_t) - x(\vec{C}|\emptyset, \emptyset)) \\ &+ \gamma_{pix}(x(\vec{C}|\emptyset, \emptyset) - x(\vec{\emptyset}|\emptyset, \emptyset)) \end{aligned} \tag{6}$$

where $\gamma_{cls}, \gamma_{typ}, \gamma_{pix}$ are guidance scales for controlling the generation. As shown in Fig. 4 (b), $x(\vec{C}|c, c_t)$ denotes the image tokens obtained by prefix $c, c_t$ and teacher forcing with $C$. $\emptyset$ denotes an empty token that avoids teacher forcing with $c, c_t$ and $C$ respectively. After obtaining the predicted tokens, the image can be decoded by a decoder as

$$I = \Phi^{-1}(x_1^*, x_2^*, \ldots, x_T^*). \tag{7}$$

For the image-to-condition generation (Fig. 4 (c)), $y^*$ can be obtained similarly by teacher forcing with $I$ and decoded similarly with the shared decoder $\Phi^{-1}$. Since teacher forcing is leveraged in the entire sampling process, we term the proposed strategy teacher-forcing guidance (TFG). More analysis of TFG is available in the Appendix.

## 4 EXPERIMENT

### 4.1 EVALUATION SETTINGS

**Dataset.** We conduct all the experiments on the ImageNet Deng et al. (2009) dataset. To incorporate pixel-level controls, we leverage state-of-the-art image understanding models to pseudo-label the images. Specifically, we label entity masks Kirillov et al. (2023), canny Canny (1986), depth Ranftl et al. (2020) and normal Vasiljevic et al. (2019) for both training and validation sets. This takes 500 Tesla V100 for about 4 days. We will release the pseudo-labeled datasets to facilitate the community to further explore conditional image generation.

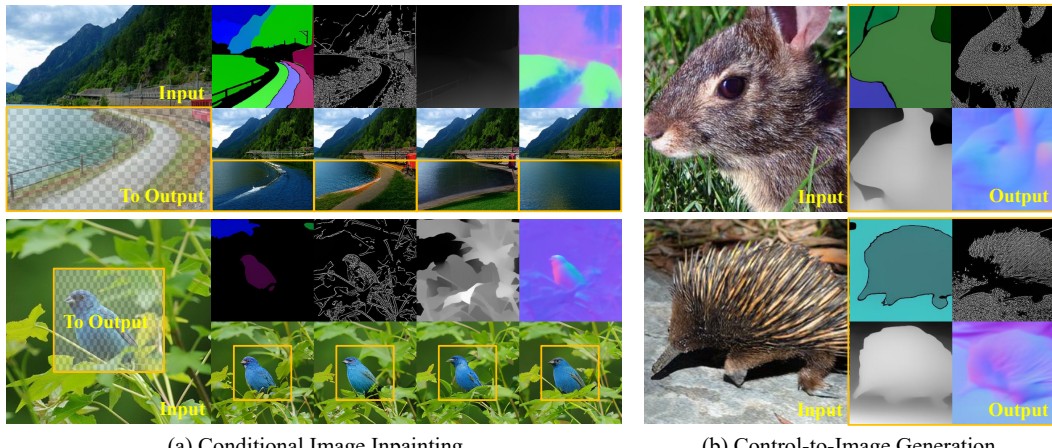

(a) Conditional Image Inpainting        (b) Control-to-Image Generation

Figure 7: Visualization of conditional image inpainting (given pixel-level control and partial image).

**Evaluation metrics.** We utilize Fréchet Inception Distance (FID) Heusel et al. (2017), Inception Score (IS) Salimans et al. (2016), Precision, and Recall as metrics for assessing the quality of image generation. However, for the image-to-control prediction where ground truth is unavailable, we rely on qualitative visualization to demonstrate the perceptual quality.

**Implementation details.** We follow VAR Tian et al. (2024) to use a GPT-2 Radford et al. (2019) style transformer with adaptive normalization Zhang et al. (2018). A transformer layer depth from 12 to 30 is explored. We leverage the pre-trained VAR tokenizer Tian et al. (2024) to tokenize both image and control. We initialize the model with the weights from VAR Tian et al. (2024) to shorten the training process. For each depth, we train the model for 30 epochs with an Adam optimizer. We follow the same learning rate and weight decay as VAR. During training, we sample each control type uniformly. To apply the classifier-free guidance, we replace class and control type conditions with empty tokens with 0.1 probability. We train the model with batchsize $= 128$ for all the experiments. During inference, we utilize top-$k$ top-$p$ sampling with $k = 900$ and $p = 0.96$. We utilize $256 \times 256$ image size for all experiments. For simplicity, we leverage $\gamma_{cls} = \gamma_{typ} = \gamma_{pix}$ for all the experiments.

## 4.2 PERFORMANCE ANALYSIS

**Joint image-control generation.** We demonstrate the performance of ControlVAR with different output control types, model sizes and guidance scales as shown in Fig. 5 (a) and Fig. 5 (b). As the model size increases, we notice ControlVAR performs better generation capability accordingly. Among all control types, jointly generating canny and image leads to a slightly inferior performance compared to other types. We consider the complex pattern of canny may impose difficulty in generating corresponding images thus leading to the degradation. In addition, we notice the optimum FID can be achieved with a guidance scale between 2 to 3. Though further increasing the guidance scale can still improve the IS, it will limit the mode diversity. We demonstrate qualitative visualization of joint generation in Fig. 6 (a) which shows high-quality and aligned image-control pairs.

Furthermore, we compare the image FID with pure image generation model VAR Tian et al. (2024) in Tab 1. We notice that ControlVAR shows a slight performance degradation compared to VAR which can be due to the difficulty enrolled to incorporate additional controls. As the model size increases, we notice the performance gap shrinks, indicating joint modeling of image and control may require more network capacity compared to image-only modeling.

| Depth | 16 | 20 | 24 | 30 |
|---|---|---|---|---|
| VAR | 3.60 | 2.95 | 2.33 | 1.97 |
| ControlVAR | 4.25 | 3.25 | 2.69 | 1.98 |

Table 1: Image FID compared to VAR.

**Conditional image generation.** We introduce two baseline methods - ControlNet Zhang et al. (2023) and T2I-Adapter Mou et al. (2023) to compare the conditional generation capability. We train

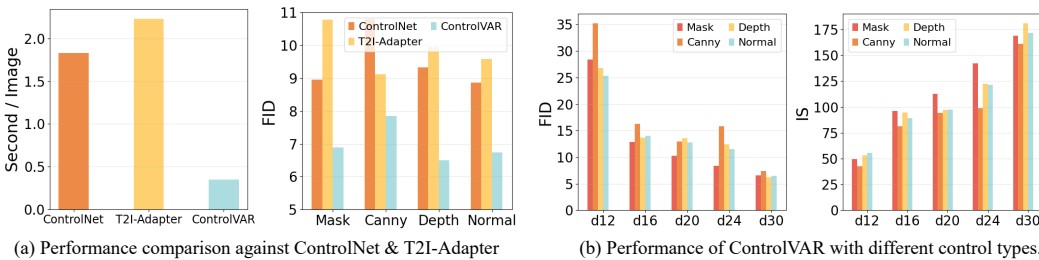

(a) Performance comparison against ControlNet & T2I-Adapter          (b) Performance of ControlVAR with different control types.

Figure 8: Quantitative results of conditional image generation.

**Guidance Scale**

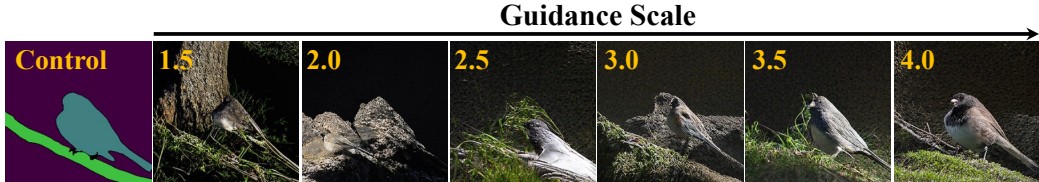

Figure 9: Visualization of images generated with different guidance scales.

both baselines on the same datasets as ours with the Diffuser von Platen et al. (2022) implementation (for a fair comparison, ImageNet-pretrained LDM Rombach et al. (2021) is used as the base model). We compare the image generation quality in terms of FID and speed in Fig. 8. We notice that ControlVAR achieves obvious superior FID compared to baselines. We evaluate the inference speed with $\mathrm{batchsize} = 1$ on a single H100 GPU. We notice that ControlVAR inference is at least 5 times faster than the compared methods. We further explore the generation capability with different model sizes as shown in Fig. 8 (b), we notice that the model's generation capability keeps improving as the model size increases. Similar to the joint image-control generation, we notice that canny-conditioned generation shows an inferior performance due to its complex pattern.

We further compare ControlVAR with conditional AR models - VQ-GAN Esser et al. (2021) and IQ-VAE Zhan et al. (2022). We fine-tune ControlVAR on ADE 20K for 1 epoch and report the FID of the generated images in Tab 2.

| Method | VQ-GAN | IQ-VAE | ControlVAR |
| --- | --- | --- | --- |
| FID | 35.5 | 29.77 | 9.72 |

Table 2: FID comparison on ADE20K.

ControlVAR demonstrates superior performance compared to previous AR methods.

**Conditional image inpainting.**    ControlVAR can support more complex image generation tasks by teacher-forcing with partial image/control. As shown in Fig. 7 (a), we showcase the conditional image inpainting results where pixel-level control and partial image are given to complete the missing part of the image. We notice that the contents align well with both the given control and image.

**Image-to-control prediction.**    ControlVAR is also capable of image understanding tasks by teacher-forcing with images during inference. As shown in Fig. 7 (b), we demonstrate the visualization of the generated controls given images. Since the pseudo labels that we use during training and inference are mediocre in quality, we do not focus on the understanding capability of ControlVAR in this paper and leave it for future work instead.

### 4.3 ABLATION EXPERIMENTS

**Module effectiveness.**    We conduct ablation experiments to validate the effectiveness of components in ControlVAR. We start with a depth 16 baseline which models the control and image in different scales without joint modeling. Tab 3 shows the impact of adding each component. We notice an obvious performance improvement by using joint modeling. Unlike the baseline setting, joint modeling enables both control and image to interact with each other on the same scale leading to better pixel-level alignment for the teacher forcing during inference. In addition, with the multi-control training and teacher forcing guidance, ControlVAR achieves 5.19 and 15.21 FID for joint

| ID | Method | Joint Control-Image | | Control-to-Image | |
|---|---|---|---|---|---|
| | | FID↓ | IS↑ | FID↓ | IS↑ |
| 1 | Baseline (w/o joint modeling) | 12.23 | 119.65 | 35.92 | 42.50 |
| 2 | + Joint modeling | $9.74_{-2.49}$ | $142.08_{+22.43}$ | $17.44_{-18.48}$ | $77.38_{+34.88}$ |
| 3 | + Multi-control training | $5.19_{-4.55}$ | $223.10_{+81.02}$ | $16.33_{-1.11}$ | $98.62_{+21.24}$ |
| 4 | + Teacher-forcing guidance | - | - | $15.21_{-1.12}$ | $95.44_{+3.18}$ |
| 5 | + Guidance scaling | $4.35_{-0.84}$ | $253.08_{+29.98}$ | $12.97_{-2.24}$ | $96.42_{+0.98}$ |
| 6 | + Larger model size | $2.09_{-2.26}$ | $337.86_{+84.78}$ | $6.57_{-6.40}$ | $173.02_{+76.6}$ |

Table 3: Ablation study on components in ControlVAR. We evaluate the FID and IS on the ImageNet validation set with masks as the target controls.

control-image and control-to-image generation respectively. During inference, we linearly anneal the guidance scale using $\gamma \cdot \frac{t}{T}$ (where $t$ is the iteration number, $T$ is the total AR iterations, and $\gamma$ is a constant hyperparameter) which brings another 0.84 and 2.24 FID gains. Lastly, by scaling the model size to depth 30, we achieve the best results of 2.09 and 6.57 FID.

**Teacher forcing guidance.** Given the same mask control, we further visualize the images generated with different guidance scales in Fig. 9. As the guidance scale increases, the generated contents align more with the given control, indicating that the TFG can effectively enhance the guidance effect.

**Generalization to unseen tasks.** As shown in Fig. 10, we conduct an unseen task by teacher-forcing a mask in the AR prediction and setting the type token to predict the canny. We notice ControlVAR can successfully generate aligned results. We optimize ControlVAR with the joint distribution between the image and controls $\sum_n p(I, C_n)$ during training which can be assumed as an alternating optimization of $p(I, \{C_n\})$. We consider this to explain the observed zero-shot capability with unseen control-to-control tasks. More visualizations are available in the Appendix.

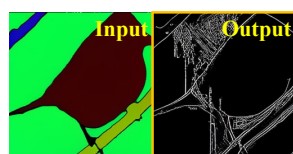

Figure 10: Mask-to-Canny.

## 5 CONCLUSION

In this paper, we present ControlVAR, an autoregressive (AR) approach for conditional generation. Unlike traditional conditional generation models that leverage prefix pixel-level controls, e.g., mask, canny, normal, and depth, ControlVAR jointly models image and control conditions during training and enables flexible conditional generation during testing by teacher forcing. Inspired by the classifier-free guidance, we introduce a teacher-forcing guidance strategy to facilitate controllable sampling. Comprehensive and systematic experiments are conducted to demonstrate the effectiveness and characteristics of ControlVAR, showcasing its superiority over powerful DMs in handling multiple conditions for diverse conditional generation tasks.

**Limitations.** In spite of ControlVAR's high performance on image generation with heterogeneous pixel-level controls, it does not support text prompts and therefore cannot be directly leveraged with natural language guidance. Developing text-guided capability can be achieved by replacing the class token with the language token, e.g., CLIP token Radford et al. (2021), which is left as our future focus.

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

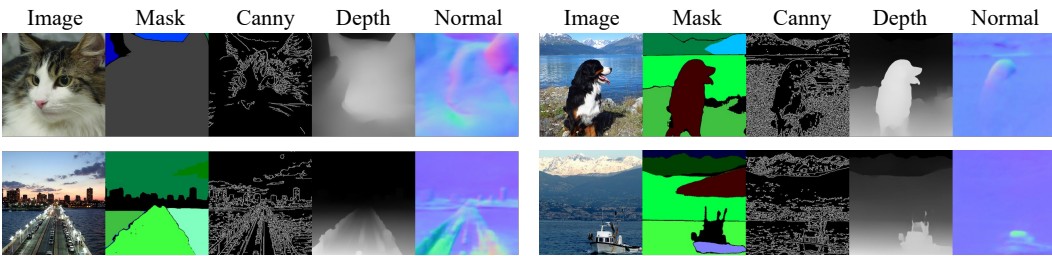

Figure A: Example of image and corresponding controls in the pseudo-labeled dataset.

## A    DATASET CREATION

We conduct all the experiments on the ImageNet Deng et al. (2009) dataset. To incorporate pixel-level controls, we leverage state-of-the-art

| Type | Mask | Canny | Depth | Normal |
|---|---|---|---|---|
| # Sample | 1277548 | 1277653 | 1277636 | 1277639 |

Table A: Statistics of the generated dataset.

image understanding models to pseudo-label the images. Specifically, we label entity masks Kirillov et al. (2023), canny Canny (1986), depth Ranftl et al. (2020) and normal Vasiljevic et al. (2019) for both training and validation sets. This takes 500 Tesla V100 for about 4 days. We demonstrate the label number after filtering in Tab A. In addition, we also manually check the quality of the pseudo labels. We show a visualization of the generated datasets in Fig. A. We notice that image understanding models predict reasonable results on ImageNet images.

## B    DISCUSSION OF THE TEACHER FORCING GUIDANCE

Inspired by the classifier-free guidance Ho & Salimans (2022) from diffusion models, we empirically find a similar form of guidance that can be used for autoregressive sample conditional images based on teacher forcing. In this section, we aim to analyze the spirit of classifier-free guidance (CFG) and analogy it to our teacher-forcing guidance (TFG).

### B.1    CLASSIFIER-FREE GUIDANCE

For the image generation task $p(I|C, c, c_t)$, given image $I$, pixel-level control $C$ and token-level controls $c_c, c_t$, CFG leverages Bayesian rule to rewrite the conditional distribution as

$$p(I|C, c, c_t) = \frac{p(c|I, C, c_t)p(c_t|I, C)p(C|I)p(I)}{p(C, c, c_t)}$$

$$\implies \log p(I|C, c, c_t) = \log p(c|I, C, c_t) + \log p(c_t|I, C) + \log p(C|I) + \log p(I) - \log p(C, c, c_t)$$

$$\implies \nabla_I \log p(I|C, c, c_t) = \nabla_I \log p(c|I, C, c_t) + \nabla_I \log p(c_t|I, C) + \nabla_I \log p(C|I) + \nabla_I \log p(I)$$

By applying the Bayesian rule again, we have

$$p(c|I, C, c_t) = \frac{p(I|c, c_t, C)p(c, c_t, C)}{p(I|c_t, C)p(c_t, C)}$$

$$\implies \nabla_I \log p(c|I, C, c_t) = \nabla_I \log p(I|c, c_t, C) - \nabla_I \log p(I|c_t, C).$$

Similarly, by applying the Bayesian rule to all terms, we have

$$\begin{aligned}
\nabla_I \log p(I|c, c_t, C) =& \nabla_I \log p(I) \\
&+ \nabla_I \log p(I|c, c_t, C) - \nabla_I \log p(I|c_t, C) \\
&+ \nabla_I \log p(I|c_t, C) - \nabla_I \log p(I|C) \\
&+ \nabla_I \log p(I|C) - \nabla_I \log p(I)
\end{aligned}$$

In the diffusion models, $\nabla_I \log p(I|*)$ is represented by the logits outputted by the diffusion-UNet. In this way, during inference, the classifier-free guidance can be calculated as

$$
\begin{aligned}
x^* =\,& x(\emptyset, \emptyset, \emptyset) \\
& + \gamma_c(x(c, c_t, C) - x(\emptyset, c_t, C)) \\
& + \gamma_{c_t}(x(\emptyset, c_t, C) - x(\emptyset, \emptyset, C)) \\
& + \gamma_C(x(\emptyset, \emptyset, C) - x(\emptyset, \emptyset, \emptyset))
\end{aligned}
$$

where $\gamma_C, \gamma_c, \gamma_{c_t}$ are the guidance scales that are used to adjust the amplitude to apply the conditional guidance. $\emptyset$ denotes leveraging a special empty token to replace the original token to disable the additional conditional information Ho & Salimans (2022).

### B.2 TEACHER FORCING GUIDANCE

Classifier-free guidance has been proven to be effective in AR models Chang et al. (2023); Tian et al. (2024) which take the same form as diffusion models as

$$
p(I|C, c, c_t) \propto p(c|I, C, c_t)p(c_t|I, C)p(C|I)p(I).
$$

In ControlVAR, we model the joint distribution of the controls and images. Therefore, we leverage a different extension of the probabilities as

$$
p(c|I, C, c_t) = \frac{p(I, C|c, c_t)p(c, c_t)}{p(I, C|c_t)p(c_t)} = \frac{p(I, C|c, c_t)p(c)}{p(I, C|c_t)}
$$

where $p(I, C|c, c_t)$ and $p(I, C|c, c_t)$ can be found from the output of ControlVAR. We follow previous works Tian et al. (2024); Esser et al. (2021) to ignore the constant probabilities $p(c)$. By rewriting all terms with Baysian rule, we have

$$
\begin{aligned}
\log p(I|C, c, c_t) \propto\,& \log p(I) \\
& + \log p(I, C|c, c_t) - \log p(I, C|c_t) \\
& + \log p(I, C|c_t) - \log p(I, C) \\
& + \log p(I, C) - \log p(I).
\end{aligned}
$$

This corresponds to the image logits as discussed in the Eq. (6)

$$
\begin{aligned}
x^* =\,& x(\uparrow\emptyset|\emptyset, \emptyset) + \gamma_{cls}(x(\uparrow C|c, c_t) - x(\uparrow C|\emptyset, c_t)) \\
& + \gamma_{typ}(x(\uparrow C|\emptyset, c_t) - x(\uparrow C|\emptyset, \emptyset)) \\
& + \gamma_{pix}(x(\uparrow C|\emptyset, \emptyset) - x(\uparrow\emptyset|\emptyset, \emptyset))
\end{aligned}
\tag{8}
$$

where $\gamma_{cls}, \gamma_{typ}, \gamma_{pix}$ are guidance scales for controlling the generation.

## C FULL RESULTS OF PERFORMANCE ANALYSIS

### C.1 DETAILS OF EVALUATION METRICS

**Fréchet Inception Distance (FID)** Heusel et al. (2017). FID measures the distance between real and generated images in the feature space of an ImageNet-1K pre-trained classifier Szegedy et al. (2016), indicating the similarity and fidelity of the generated images to real images.

**Inception Score (IS)** Salimans et al. (2016). IS also measures the fidelity and diversity of generated images. It consists of two parts: the first part measures whether each image belongs confidently to a single class of an ImageNet-1K pre-trained image classifier Szegedy et al. (2016) and the second part measures how well the generated images capture diverse classes.

**Precision and Recall** Kynkäänniemi et al. (2019). The real and generated images are first converted to non-parametric representations of the manifolds using k-nearest neighbors, on which the Precision and Recall can be computed. Precision is the probability that a randomly generated image from estimated generated data manifolds falls within the support of the manifolds of estimated real data distribution. Recall is the probability that a random real image falls within the support of generated data manifolds. Thus, precision measures the general quality and fidelity of the generated images, and recall measures the coverage and diversity of the generated images.

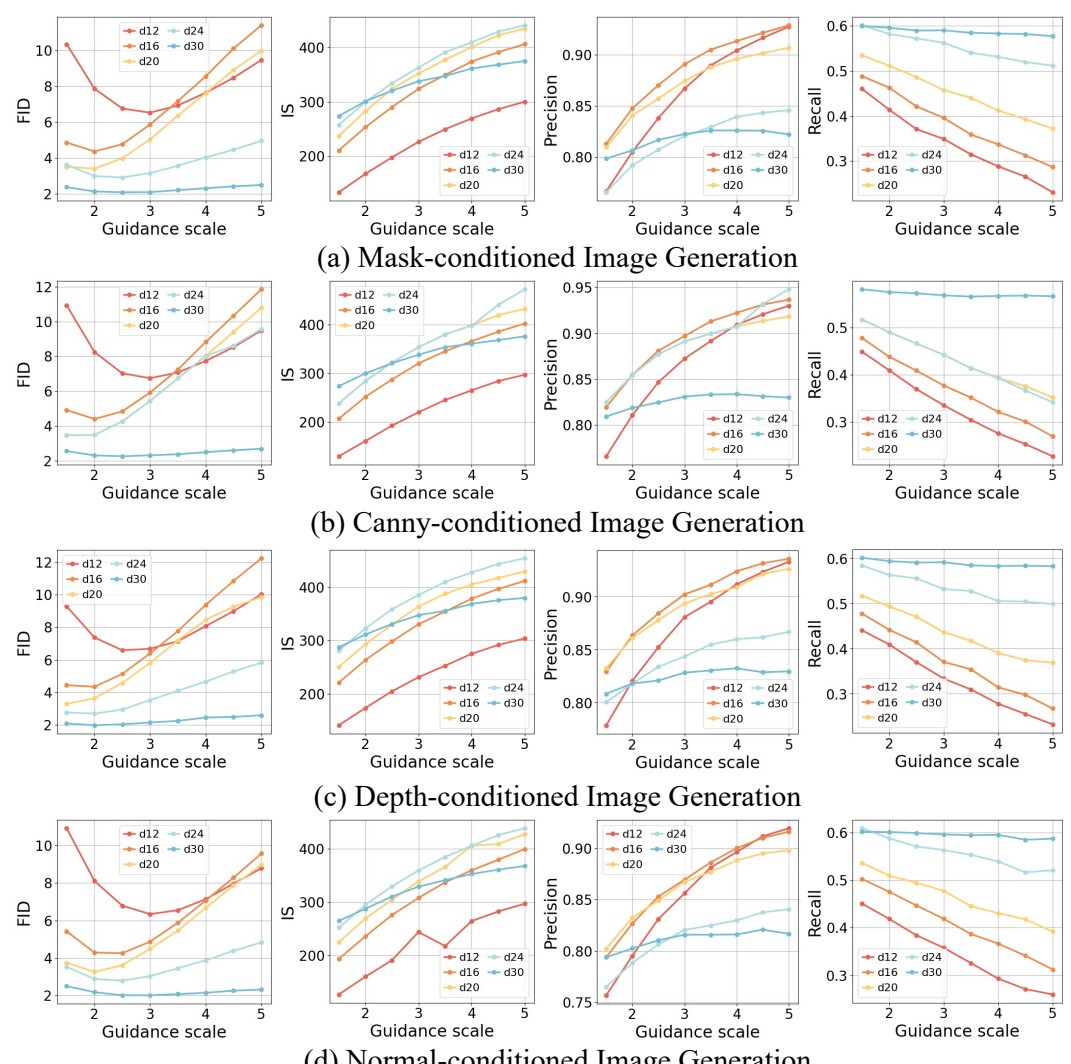

Figure B: Performance of joint image-control generation for different control types. The performance is evaluated on the ImageNet validation set with our created pseudo labels.

## C.2  JOINT CONTROL-IMAGE GENERATION

We demonstrate more detailed results for each control type for joint control-image generation in Fig. B. As the model size increases, ControlVAR demonstrates better performance. The control generated along with the image shows a minor impact on the image quality.

## C.3  CONTROL-TO-IMAGE GENERATION

We demonstrate more results when comparing with baseline models - ControlNet and T2I-Adapter in Fig. C and Fig. I. The performance is evaluated on the ImageNet validation set with our created pseudo labels. We notice that ControlVAR demonstrates a superior performance for different tasks. Specifically, we notice that ControlNet outperforms ControlVAR for canny-conditioned image generation. We consider this to be due to the difficulty of handling the joint modeling of the complex canny and image. In addition, we notice that when the guidance scale increases, ControlNet and T2I-Adapter demonstrate a superior inception score and an inferior FID, we consider this attributed to the increasing mode collapse resulting from the larger guidance scale. In contrast, the performance of ControlVAR is more robust.

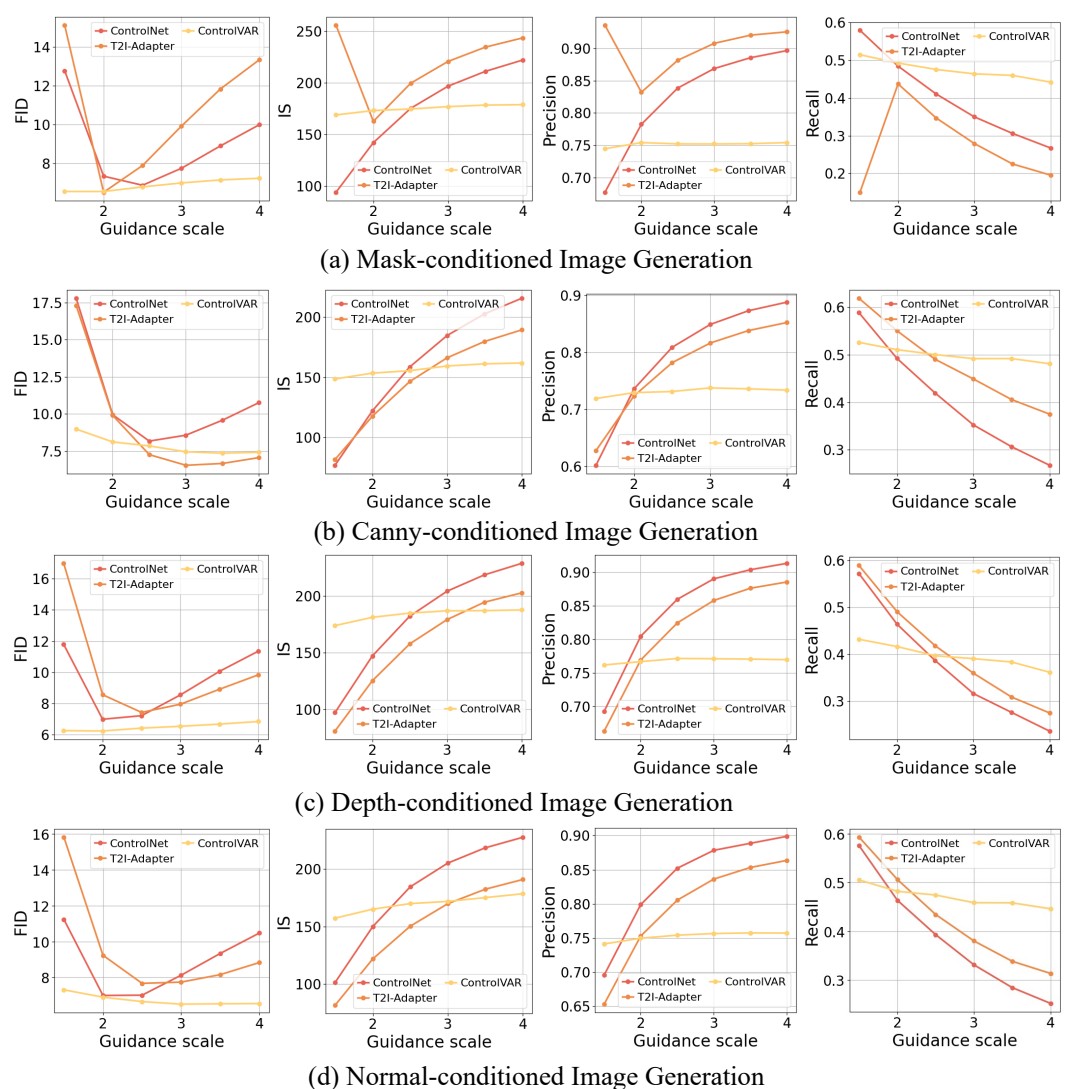

(a) Mask-conditioned Image Generation

(b) Canny-conditioned Image Generation

(c) Depth-conditioned Image Generation

(d) Normal-conditioned Image Generation

Figure C: Performance of conditional generation for different condition types. We first introduce two baseline methods - ControlNet Zhang et al. (2023) and T2I-Adapter Mou et al. (2023) to compare the conditional generation capability. We train both baselines on the same datasets as ours with the Diffuser von Platen et al. (2022) implementation (for a fair comparison, ImageNet-pretrained LDM Rombach et al. (2021) is used as the base model). The performance is evaluated on the ImageNet validation set with our created pseudo labels.

## D    MORE VISUALIZATION

We demonstrate more qualitative visualization for joint control-image generation (Fig. F), image/control completion (Fig. G), image perception (Fig. H), conditional image generation (Fig. I) and unseen control-to-control generation (Fig. E).

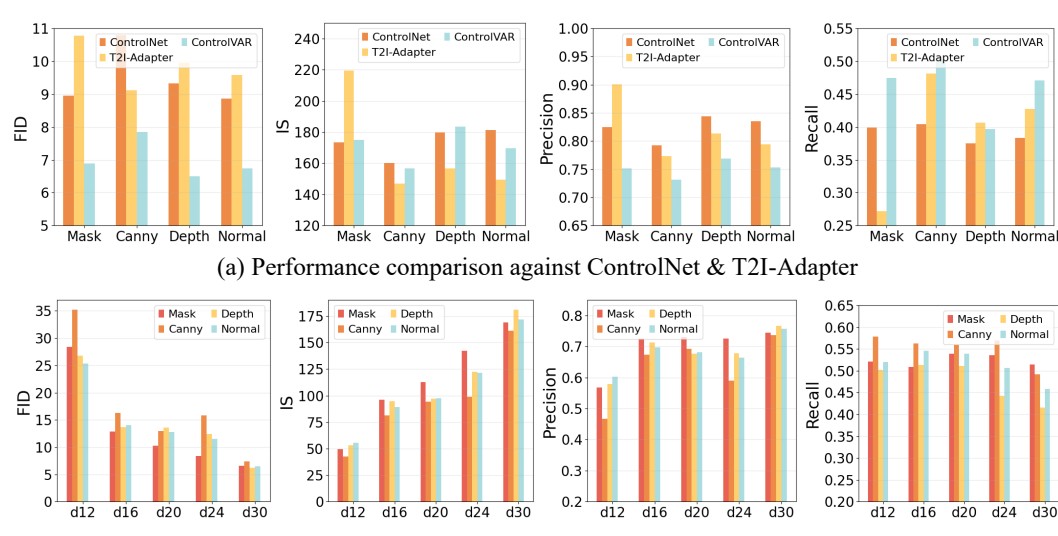

(a) Performance comparison against ControlNet & T2I-Adapter

(b) Performance of ControlVAR with different control types and model sizes.

Figure D: Performance of conditional generation for different condition types. We first introduce two baseline methods - ControlNet Zhang et al. (2023) and T2I-Adapter Mou et al. (2023) to compare the conditional generation capability. We train both baselines on the same datasets as ours with the Diffuser von Platen et al. (2022) implementation (for a fair comparison, ImageNet-pretrained LDM Rombach et al. (2021) is used as the base model). The performance is evaluated on the ImageNet validation set with our created pseudo labels.

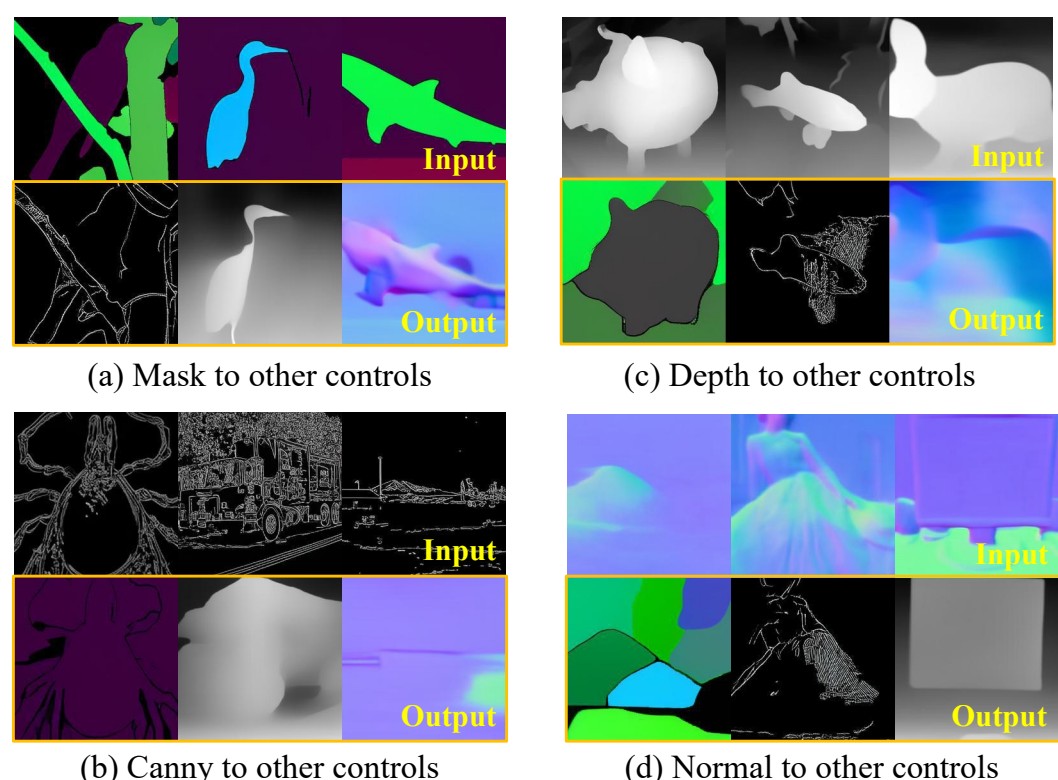

(a) Mask to other controls

(c) Depth to other controls

(b) Canny to other controls

(d) Normal to other controls

Figure E: Qualitative visualization for zero-shot condition understanding task. The yellow boxes denote the predicted controls.

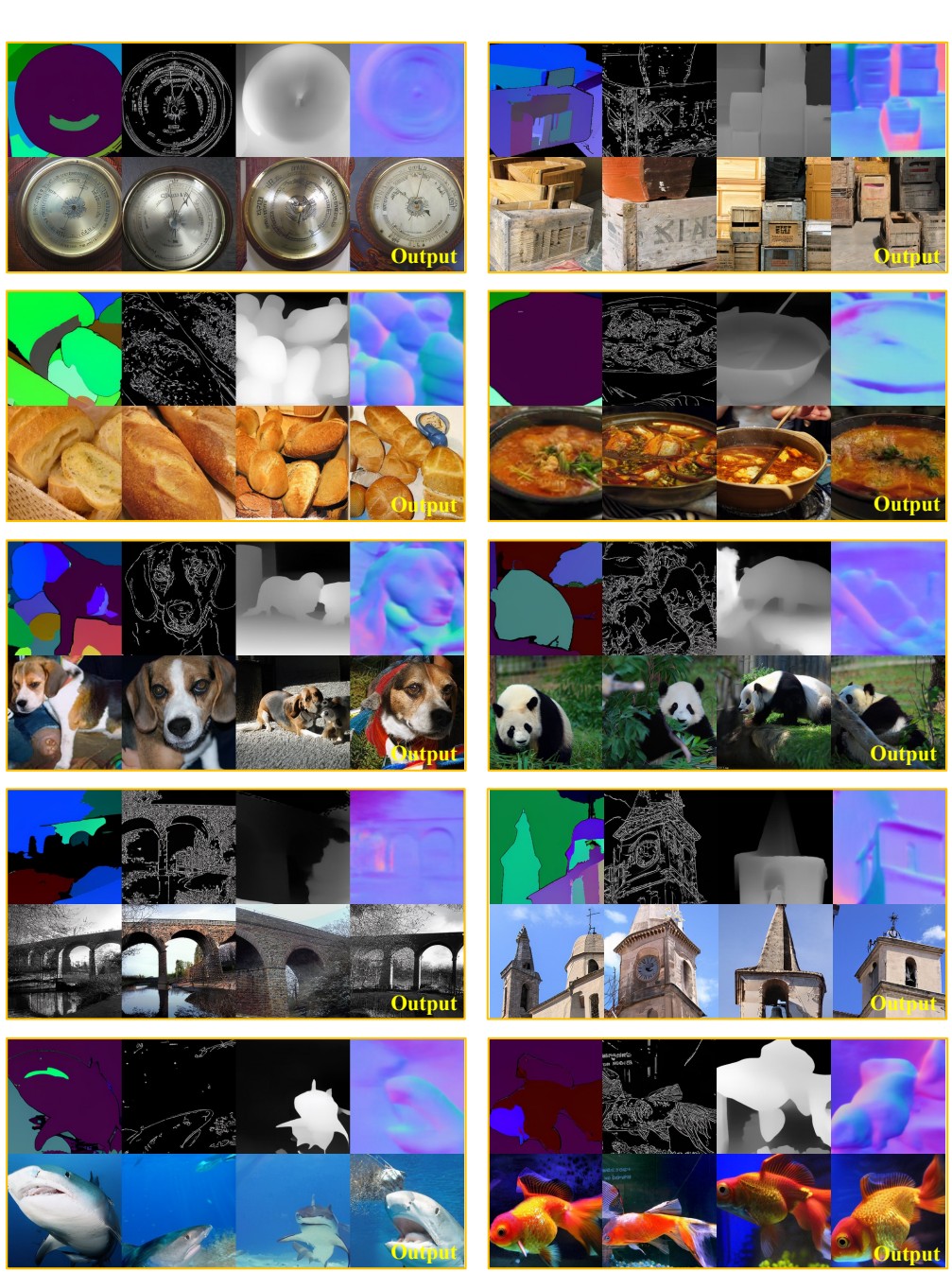

Figure F: Qualitative visualization for joint control-image generation task. The yellow boxes denote the predicted images & controls.

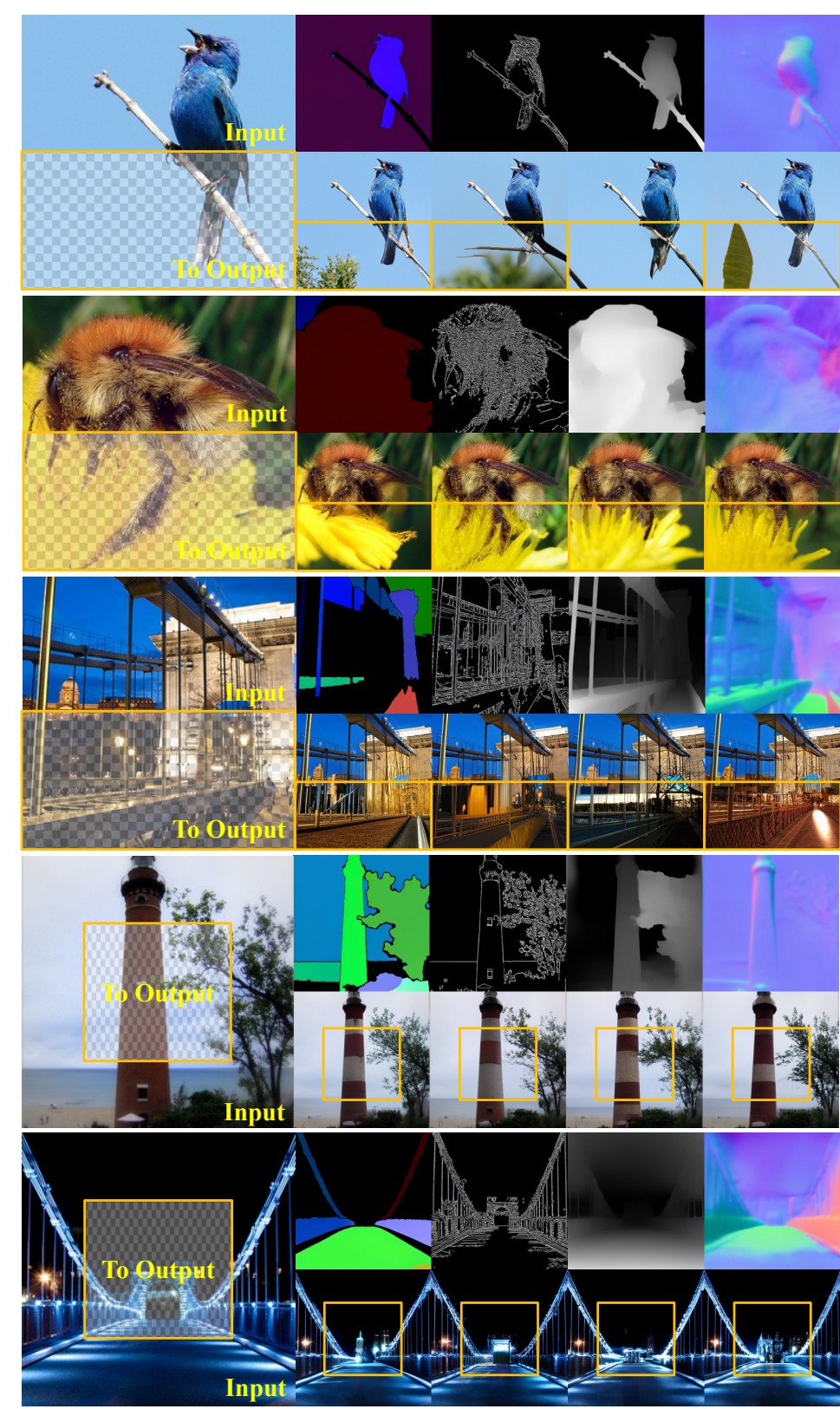

Figure G: Qualitative visualization for image/control inpainting task. The yellow boxes denote the predicted images/controls.

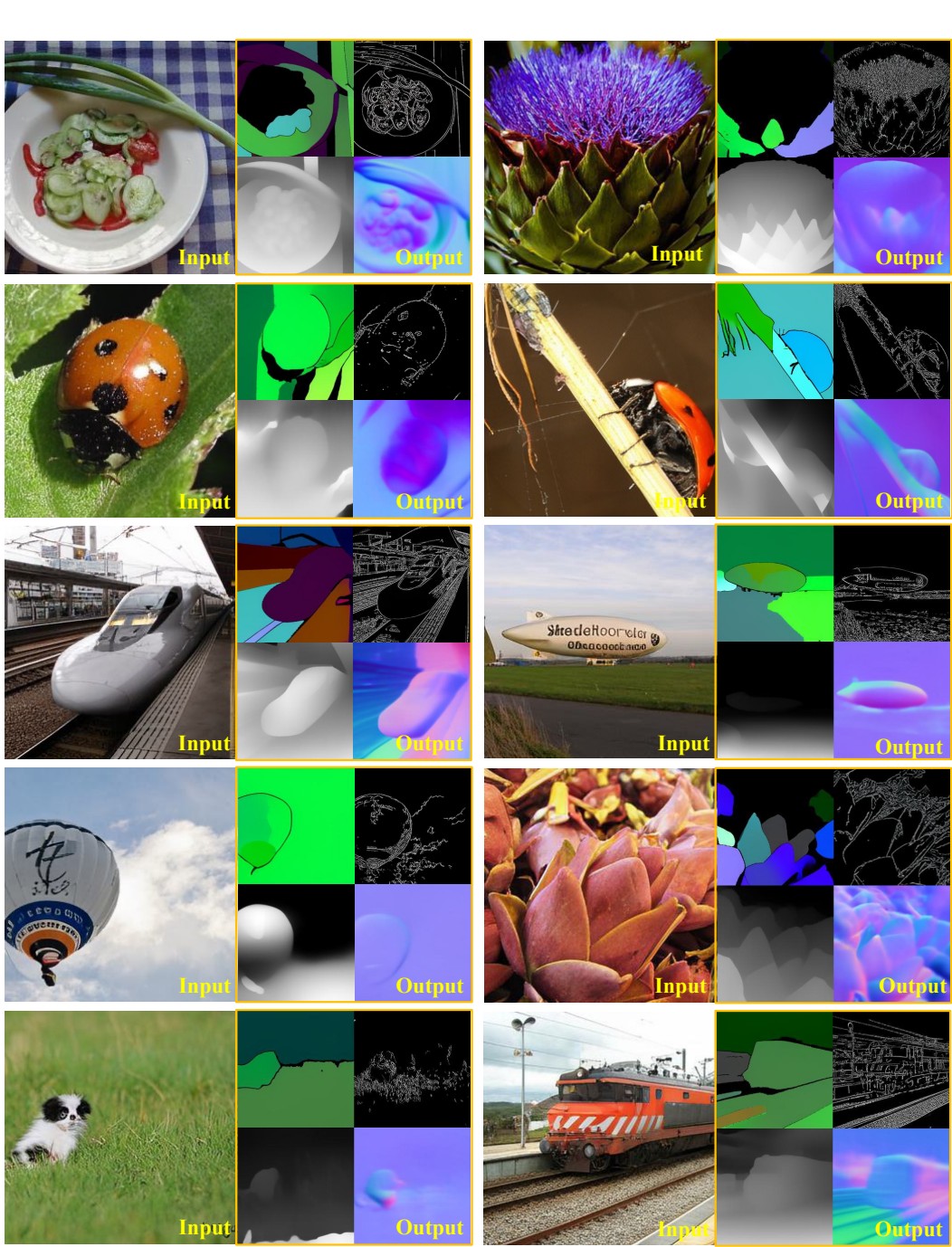

Figure H: Qualitative visualization for image understanding task. The yellow boxes denote the predicted controls.

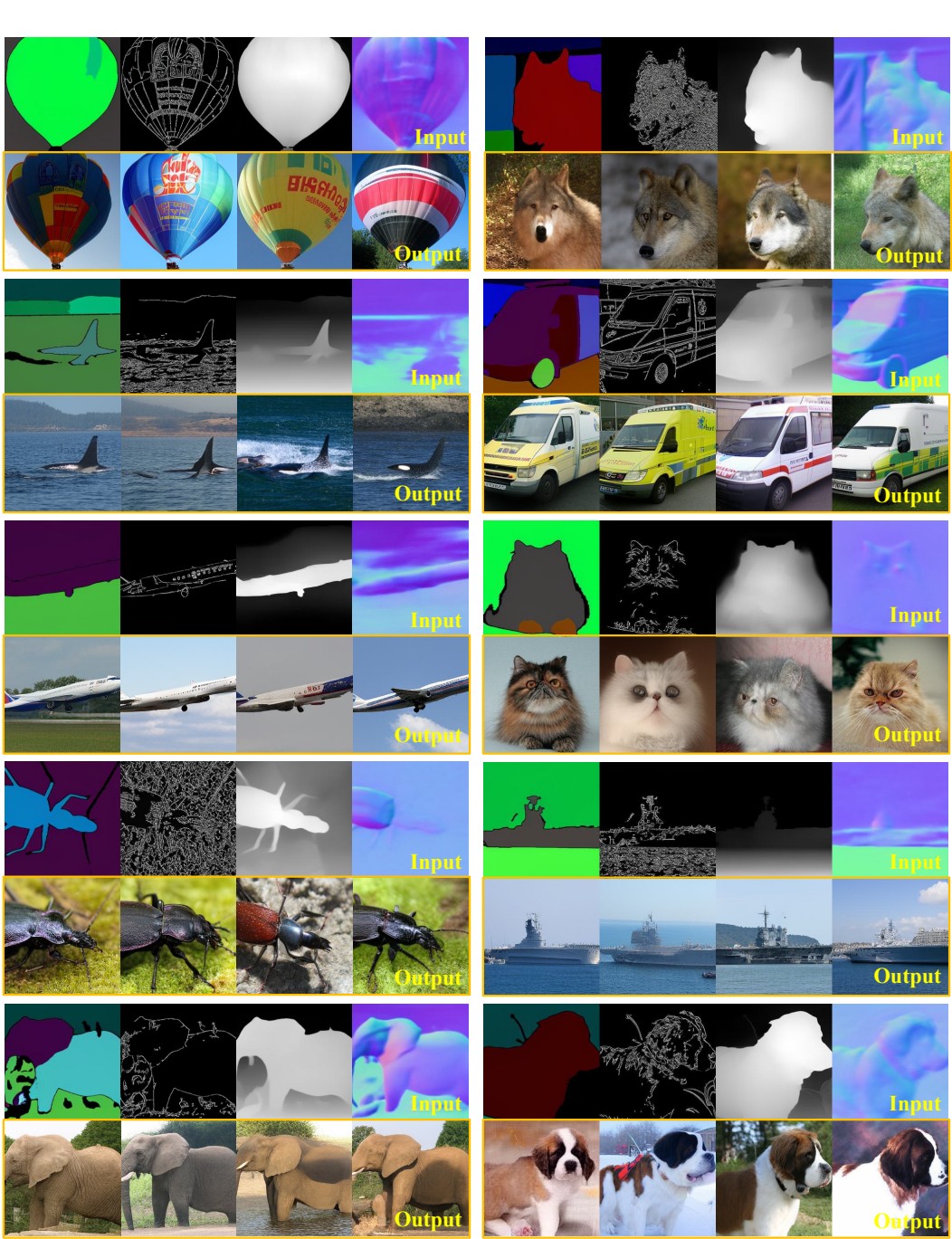

Figure I: Qualitative visualization for conditional image generation task. The yellow boxes denote the predicted images.

