# OpenReview forum: "ControlVAR: Exploring Controllable Visual Autoregressive Modeling"
_ICLR.cc/2025/Conference — ICLR 2025 Conference Withdrawn Submission_

### Official Review · Reviewer_w4Su · 2024-11-01

**Soundness:** 2
**Presentation:** 2
**Contribution:** 2
**Rating:** 5
**Confidence:** 5

**Summary:**

This paper introduces ControlVAR, an autoregressive framework for conditional image generation that jointly models images and pixel-level controls, enabling flexible and efficient visual generation. By employing a teacher-forcing guidance strategy, ControlVAR facilitates controlled testing across various tasks, including control-image generation, inpainting, and image-to-control prediction. Experimental results indicate that ControlVAR performs competitively with diffusion-based models, such as ControlNet, suggesting its suitability as an alternative for controllable visual generation with potentially lower computational costs.

**Strengths:**

1.	This paper focuses on controllable image generation using autoregressive models, a forward-looking area with substantial application potential, providing valuable insights for the research community.
2.	The paper introduces ControlVAR, which employs pixel-level controls in autoregressive modeling for controllable image generation, and employs several innovative mechanisms, such as teacher-forcing guidance (TFG) for controllable sampling.
3.	The paper provides some empirical results, outperforming popular DMs, such as ControlNet and T2I-Adapter, across different conditional generation tasks.

**Weaknesses:**

1.	ControlVAR requires tuning the pre-trained VAR model, which limits the flexibility of the proposed method. Switching to a new base model still requires retraining, making this approach resource-intensive. In diffusion models like Stable Diffusion, ControlNet does not modify SD’s weights. However, if there is a mature autoregressive generation model with a parameter size similar to SD in the future, ControlVAR would require retraining or fine-tuning it, which would be unacceptable in most application scenarios.
2.	In Section 3.1, the author provides extensive details on a particular method of control extraction, but this content may not be suitable to occupy a significant portion of the core methodology section; it might be more appropriate to place it in the experiments section or supplementary materials for a more detailed discussion.
3.	Similarly, the Tokenization part merely states that the same tokenizer as VAR is used, offering minimal informative value.
4.	The concept of methods is only introduced in Section 3.2, resulting in a lack of detailed information throughout the methods section, and the interpretation is not sufficiently in-depth. The overall network modeling section could be described in a more specific and theoretically grounded manner.

**Questions:**

Although the field of controllable image generation using autoregressive models is novel, and the experiments showcase the strong controllability of the proposed ControlVAR, the main concern for reviewer lies in the method’s scalability to other autoregressive models. If a much larger and more powerful base model emerges in the future, would it also require fine-tuning to achieve controllability? (ControlNet, after all, doesn’t require retraining Stable Diffusion.) Perhaps freezing the base model could be a solution worth exploring, and we would appreciate the authors’ perspective on this point.

---

### Official Review · Reviewer_HmTw · 2024-11-03

**Soundness:** 2
**Presentation:** 2
**Contribution:** 1
**Rating:** 3
**Confidence:** 5

**Summary:**

The paper introduces **ControlVAR**, a “controllable autoregressive” (AR) image generation framework designed to enhance flexibility and efficiency in conditional generation tasks, particularly as an alternative to diffusion models (DMs) due to their computational costs and latency.

The work jointly models image and pixel-level conditions, using a teacher-forcing guidance (TFG) strategy that improves controllable generation by substituting predicted values with ground truth during inference.

This approach enables adapted VAR to handle a wide range of tasks—such as control-to-image and image-to-control generation—and even unseen tasks like control-to-control generation.

The work demonstrates good performance, compared with ControlNet and T2I-Adapter across multiple pixel-level control tasks, including mask, canny, depth, and normal control.

**Strengths:**

1. **Promising Direction for AR Model Control**: The paper addresses an important and interesting challenge—how to control autoregressive (AR) models effectively—which is valuable for the community as AR applications grow.
2. **Well-Designed Experiments**: The experiments are thoughtfully set up for various tasks, providing a clear view of the framework’s capabilities and its performance compared to popular models.
3. **Clear Writing**: The paper is well-written and easy to follow, making the technical aspects and contributions understandable to a broad audience.

**Weaknesses:**

1. **Resource-Intensive Tuning and Limited Flexibility**: My major concern is the motivation for the work. The limitation of ControlVAR is its requirement for fine-tuning the pre-trained VAR model, which reduces the method’s flexibility and scalability. Unlike diffusion models such as Stable Diffusion, where ControlNet adds control without altering the base model’s weights, ControlVAR necessitates modifications to the underlying VAR model to enable control. This limitation makes it less practical for applications that demand flexibility across diverse base models, as switching to a new foundational model would still require retraining or extensive fine-tuning, resulting in substantial computational and resource overhead.
2. **Lack of Adaptability**: ControlVAR’s requirement for retraining or fine-tuning when switching base models significantly limits its adaptability, particularly in settings where retraining large models is impractical. This limitation could hinder ControlVAR's adoption in real-world applications, as it restricts the flexibility needed for various scenarios. Furthermore, ControlVAR's dependence on control conditions makes it challenging to function independently of them, reducing its suitability for tasks that might require unconditioned generation. Although extensive experiments validate the framework's performance, the primary motivation for ControlVAR should be clarified further to avoid potential confusion within the community.
3. **Delayed Method Introduction and Lack of Theoretical Depth**: The core methodology of ControlVAR is not introduced until Section 3.2, and even then, the theoretical grounding is somewhat shallow. This late introduction and limited depth in the explanation of controllable modeling make it challenging for readers to fully understand the framework’s design and motivations. The paper’s presentation could be strengthened by expanding the network modeling section with more rigorous and specific theoretical insights. A more structured approach to explaining the controllability mechanism would enhance the clarity of ControlVAR’s contributions and make the work more accessible to a broader audience. Without a thorough explanation, the framework may appear conceptually fragmented, and readers may struggle to appreciate the full scope of its methodological innovations.
4. **Limited Novelty in Teacher-Forcing Guidance**: While teacher-forcing guidance is a core element of the ControlVAR framework, it does not introduce substantial novelty and appears to be similar to existing guidance methods, such as classifier-free guidance. Its application within ControlVAR is not notably distinct from previous uses in controllable generation, which may lessen its perceived value and impact within the paper. This similarity raises questions about the uniqueness of the guidance method as it relates to enhancing control in AR models. Without a more innovative or tailored approach, teacher-forcing guidance may come across as a standard technique rather than a breakthrough for controllable generation.
5. **Focus and Theme Ambiguity**: The paper’s inclusion of various tasks, such as image-to-control and control-to-control predictions, diverges from its primary focus on control-to-image generation. This range of tasks blurs the paper’s central theme, making it challenging for readers to pinpoint the core contribution and focus of ControlVAR. While demonstrating versatility is valuable, the inclusion of these peripheral tasks risks diluting the main message and may confuse readers about the intended purpose of the model. Furthermore, the name “ControlVAR” suggests a potentially misleading emphasis on control-focused, multi-task learning, which may not align with the actual scope and main objectives of the framework. This ambiguity in the framework’s focus and naming could hinder its positioning within the broader field of controllable generation.

**Questions:**

1. ControlVAR currently requires fine-tuning when changing base models, which could limit its adaptability. Are there plans or ongoing efforts to reduce the dependency on model-specific tuning to improve scalability, particularly for scenarios where retraining large models is not feasible?

2. The theoretical framework for achieving control, introduced in Section 3.2, could be expanded to provide more depth. Could you elaborate on how control is achieved within the AR setup, particularly at the pixel level, and what makes this approach effective?

3. ControlVAR’s reliance on control conditions might limit its use in scenarios that require unconditioned generation. Is there a potential for adapting or extending ControlVAR to handle unconditioned tasks, and if so, could you describe how that might be approached?

---

### Official Review · Reviewer_CFs5 · 2024-11-05

**Soundness:** 3
**Presentation:** 1
**Contribution:** 3
**Rating:** 5
**Confidence:** 4

**Summary:**

The paper introduces ControlVAR, a novel framework for controllable visual autoregressive modeling, which integrates pixel-level controls to enhance conditional image generation. By transferring pixel-level controls like mask into the same RGB space like images, the control and the image can be tokenized using the same approach. The authors also leverage teacher-forcing guidance to enhance the sampling quality. Extensive experiments demonstrate the effectiveness of the proposed approach.

**Strengths:**

* The alternating prediction of image tokens and control tokens seems new.
* Experiments show the effectiveness of the proposed methods.
* The visualization is clear and helps illustrates the framework of the proposed method and the generation results.

**Weaknesses:**

* The organization of the paper lacks clarity, with some confusing aspects. For instance, at the beginning of Section 3, the notation is unclear: $C$ represents pixel-level control, while $c$ stands for token-level control, but the distinction between these two is not fully explained. Additionally, the problem formulation is introduced without any examples, making it challenging to follow. The first example of control only appears on page five, and the tokenization method is explained on page six. Before this point, it’s unclear why the number of control tokens must match the image tokens. Including examples or preliminaries earlier in the section would help readers understand these concepts. Moreover, the notation in Equation (6) is confusing—it's not immediately apparent how $x$ and Equation (6) are derived.
* This approach also seems computationally intensive, as ControlVAR sequences are twice as long as those in VAR, leading to at least double the training and inference time. Although the authors have compared the training speed of ControlVAR with ControlNet and T2I-Adapter, the model configurations for the comparison are not provided, which makes the comparison unconvincing.
* The citation format should be revised. Instead of relying solely on \cite, please use \citep or \citet as appropriate for the context.

**Questions:**

In addition to the questions in the weaknesses part, I have the following questions:
* The authors explore different model depths, which I appreciate; however, I am curious about the necessity of this exploration. What specific insights or conclusions can be drawn from varying the model depth?
* The competition between autoregressive and diffusion models remains strong. I would like to know the authors' perspective on the advantages of using an autoregressive approach for conditional generation compared to the more mature diffusion methods.

---

### Note · Authors · 2024-11-13

**Comment:**

I appreciate the reviewer's and AC's time and effort. I decide to withdraw the submission at this step.

**Withdrawal Confirmation:**

I have read and agree with the venue's withdrawal policy on behalf of myself and my co-authors.